# WHEN WILL FEDERATED LEARNING TRANSFER FROM GENERALIZATION TO PERSONALIZATION?

**Wenxuan Liu, Lukun Wang**
Department of Computer Technology
Shandong University of Science and Technology
{liuwenxuan,wanglukun}@sdust.edu.cn

**Jiaming Pei**
School of Computer Science
The University of Sydney
jpei0906@uni.sydney.edu.au

## ABSTRACT

The timing of personalization refers to determining when to train and update personalized models for the participants in personalized federated learning. Determining the timing for personalization contributes to improving the overall efficiency of federated learning. We propose that training transfers to personalization when the accuracy of the global model reaches a predefined threshold. Experimental results show that this method can effectively improve the accuracy of personalized models in a non-IID scenario.

## 1 INTRODUCTION

Federated learning can leverage multi-party collaboration to train models while protecting user privacy. This ensures data privacy and security while solving the problem of data silos. However, the actual application environment of federated learning is complex and heterogeneous, and various factors can affect the accuracy of the federated learning model. Traditional federated learning has slow convergence and low accuracy on highly heterogeneous data (non-IID) Kairouz et al. (2021). Moreover, a single global model is difficult to meet the needs of all clients. To address this challenge, personalized federated learning aims to provide each user with a suitable personalized model Chen et al. (2018);Wang et al. (2019);Li & Wang (2019);Arivazhagan et al. (2019);Fallah et al. (2020);Mansour et al. (2020);Deng et al. (2020);T Dinh et al. (2020). These studies prove that federated learning is necessary to transfer from generalization to personalization.

Although there are many studies on personalized federated learning, a paper that proposes when federated learning should transfer from generalization to personalization has not been found. The initial state of federated learning is always a random initial model. However, if the model is personalized in the initial state, the result will be far from the original intention of federated learning. This approach will limit the generalization of the global model and cannot fully utilize the data of all clientsKirkpatrick et al. (2017). On the other hand, if the personalization operation is performed after the global model converges, the final model of each client is misled by the global model, resulting in a suboptimal model. This is because only the accuracy of the global model is optimized, and the optimal global model obtained is not suitable for subsequent personalized processing Jiang et al. (2019). However, most of the existing studies directly use the converged global model for personalization. The time to transfer to personalization should neither cause important parameters of the global model to be changed, nor allow each client's final model to be misled by the global model.

In this work, we propose a method to determine when to transfer to personalization based on the accuracy of the global model. We set a threshold for the accuracy. When the accuracy reaches the threshold, the training transfers from generalization to personalization. The experiments show that this method can improve the accuracy of the personalized model in non-IID scenarios.

## 2 IMPLEMENTATION

In this section, we propose our method in Algorithim 1. We run the traditional FedAvgMcmahan et al. (2016) algorithm to train the global model, using SGD with momentum as the optimizer. In

| **Algorithim 1** |
| --- |
| 1: **Generalization:** run FedAvg with momentum SGD as server optimizer to train a global model |
| 2: **Transfer** to personalization when the accuracy of the global model reaches the threshold |
| 3: **Personalization:** each client conducts personalization with its own data |

the training of the global model, we focus on its accuracy and set a threshold for the accuracy. When the accuracy reaches the threshold, the training transfers from generalization to personalization. Therefore, we need to find out how many communication rounds are needed for the global model to meet the threshold and how the accuracy changes with the number of rounds. After that, each client personalizes the global model based on its own data. We adopt the same optimizer as the training of the global model, and each client builds a personalized model by gradient descent on its own data.

## 3 EXPERIMENTS AND RESULTS

In this section, we conducted experiments to examine how the personalization start time affects the personalization performance under non-IId data. We used MNIST as benchmark dataset in our study. The MNIST data set was sampled according to the Dirichlet distribution to obtain the non-IID data set. We used the symbol $Dir(\alpha)$ to represent different Dirichlet distributions controlled by $\alpha$. The smaller the value of $\alpha$, the higher the heterogeneity of the data. For this experiment, we chose $\alpha$= 0.5. We used the same CNN architecture as FedAvg and the SGD optimizer with a learning rate of 0.01. We set the SGD momentum to 0.9. The local batch size was set to B=50, and the local epoch was set to E=5. In this experiment, we had one central server and 100 clients.

Table 1: Local average accuracy

| Methods | Accuracy(%) |
| --- | --- |
| Our approach | 90.62 |
| FedAvg | 86.83 |
| FedProx | 89.25 |
| SCAFFOLD | 82.52 |

To prove the effectiveness of our proposed method, we analyzed the performance of the personalized model. We evaluated the performance of the personalized model using the average value of the local model's performance on each online client. We used FedAvg, as well as the personalized algorithms FedProxLi et al. (2018) and SCAFFOLDKarimireddy et al. (2019), as the baseline algorithms. Based on prior knowledge, the FedAvg algorithm converged in the 56th round of the MNIST data set. Therefore, we chose 40-50 rounds to start steering for personalized operation. We used FedAvg to train a global model for 40 rounds of communication and distributed it to each client as an initial model. We used client data for personalized training and compared the accuracy with the baseline algorithms.

The experimental results shown in Table 1 demonstrated that the performance of our proposed method was improved compared with the baseline algorithms. This means that our method provided better personalized performance, avoided the misleading of the global model, and obtained reliable knowledge from the global model.

## 4 CONCLUSIONS AND FUTURE WORK

Our work aims to improve the accuracy of personalized federated learning models. We propose setting a threshold for the accuracy of the global model. When the accuracy of the global model reaches the threshold, the training will transfer from generalization to personalization. Our experiments show that starting personalization before the global model converges can improve the accuracy of the personalized models. However, our current experiments are based on prior knowledge to select specific communication rounds. In future work, we will further study how to determine the optimal threshold and which clients need personalization.

URM STATEMENT

The authors acknowledge that at least one key author of this work meets the URM criteria of ICLR 2023 Tiny Papers Track.

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
