# OpenReview forum: "When will federated learning transfer from generalization to personalization?"
_ICLR.cc/2023/TinyPapers — Submitted to Tiny Papers @ ICLR 2023_

### Official Review · Reviewer_5Vou · 2023-03-24

**Confidence:** 4

**Summary Of Contributions:**

The paper proposes a personalized federated learning algorithm which aims to develop the client specific model contrary to only learning a single model applicable to each client.

**Rating:**

Great Start (GS): a submission which meets some of the reviewing criteria but has room for improvement

**Strengths And Weaknesses:**

The idea of personalization is important. A single global model might not be an ideal model for participating clients.

The idea presented is not well described. The parameters used in the experiments are missing. For example, how many clients, how the threshold has been calculated, whether is there a single threshold or multiple thresholds, what extra computational cost this thresholding will bring, etc

**Suggested Changes:**

The paper needs a detailed experimental setup to justify whether the experiments performed in the paper are fair or not.

The experiments need to be performed with various parameters such as $\alpha$

The paper lacks a literature review and its limitations. This is not the first work tackling the issue of personalization; therefore, it is important to highlight the weakness of the current literature. Further, the comparison with existing algorithms is also needed working in the personalization space.

---

> ### Author Response · Authors · 2023-05-31
> **Response to Reviewer 5Vou**
>
> Thank you very much for your valuable comments and advice. They are all very helpful for us to revise and improve our paper. The responses to your specific comments are as follows:
> >Summary Of Contributions: The paper proposes a personalized federated learning algorithm which aims to develop the client specific model contrary to only learning a single model applicable to each client.
>
> >Comment: The idea of personalization is important. A single global model might not be an ideal model for participating clients. The idea presented is not well described. The parameters used in the experiments are missing. For example, how many clients, how the threshold has been calculated, whether is there a single threshold or multiple thresholds, what extra computational cost this thresholding will bring, etc
>
> Response: Thank you for your summary and suggestion. We have included a more detailed description of the proposed method to improve the clarity of the manuscript. And Table Algorithm 1 has been added to explain the flow of our proposed method.
>
> >Comment: The paper needs a detailed experimental setup to justify whether the experiments performed in the paper are fair or not. The experiments need to be performed with various parameters such as $\alpha$.
>
> Response: Thank you very much for this comment. We present our experimental procedure in the Experiments and Results section. Our experimental parameters are as follows:
> * Dataset: MNIST
> * Non-IID parameter of Dirichlet distribution: $\alpha$=0.5
> * Number of Clients: 100
> * Number of servers: 1
> * local training epoch: 5
> * local training batch size: 50
> * SGD Momentum: 0.9
> * Learning rate: 0.01
>
> >Comment: The paper lacks a literature review and its limitations. This is not the first work tackling the issue of personalization; therefore, it is important to highlight the weakness of the current literature. Further, the comparison with existing algorithms is also needed working in the personalization space.
>
> Response: Thank you for your precious comments and advice.  We have conducted a more thorough literature review and illustrated current research gaps with examples. In the experimental section, we have sampled the MNIST dataset using the Dirichlet distribution to obtain non-IID data, and chosen the personalized algorithm FedProx and SCAFFOLD to compare with our proposed algorithm.
>
> Thank you for your careful review. We really appreciate your efforts in reviewing our manuscript.

---

> > ### Comment · Reviewer_5Vou · 2023-05-31
> > **Questions**
> >
> > 1. The proposed algorithm is written in a superficial manner. In my opinion, the authors can remove Algo-1 and can add relevant details. Please correct me if I am wrong, but are you training a global model, and once it yields an accuracy controlled by the threshold parameter, we will stop updating the global model? Further, each client will keep updating their local copy of the model to get a personalized model.
> >
> > 2. What is the performance of the personalized and global models on heterogenous and out-of-distribution datasets?
> >
> > 3. What is the performance of this approach on any complex dataset such as CIFAR-100 and its comparison with SOTA works?
> >
> > Thanks for submitting the response.

---

> > > ### Author Response · Authors · 2023-06-06
> > > **Response to Reviewer 5Vou**
> > >
> > > Thank you for your question and please accept our apologies for the delayed response.
> > > > Q: The proposed algorithm is written in a superficial manner. In my opinion, the authors can remove Algo-1 and can add relevant details. Please correct me if I am wrong, but are you training a global model, and once it yields an accuracy controlled by the threshold parameter, we will stop updating the global model? Further, each client will keep updating their local copy of the model to get a personalized model.
> > >
> > > Response: We apologize that our Algo-1 is too simple. Our original intention was to make it easier for readers to understand. We appreciate your comments and we will add more details in the next version. Yes, our method is to first train a global model that leverages the data from all clients. Once the accuracy of the trained global model reaches the threshold, each client uses its own data to fine-tune the received global model to obtain a personalized model.
> > >
> > > >Q: What is the performance of the personalized and global models on heterogenous and out-of-distribution datasets?
> > >
> > > Response: For our experiments, we used the Dirichlet distribution to sample the MNIST dataset and obtain non-IID data, that is, heterogeneous data. The global model’s accuracy decreased on non-IID data, while the personalized model’s accuracy improved compared to the global model. We apologize for not conducting experiments with OOD data yet, and we plan to do so in our future work.
> > >
> > > >Q: What is the performance of this approach on any complex dataset such as CIFAR-100 and its comparison with SOTA works?
> > >
> > > Response: Thank you for your question. We are working on expanding our experimental setup to include other datasets,  and we regret that our experiments are not perfect at this stage. We compared our method with FedProx and SCAFFOLD, which are the most commonly used personalized federated learning algorithms, and our method outperformed them on non-IID data. We will conduct more experiments in the next work to further prove the advantages of our method.

---

### Official Review · Reviewer_uWdB · 2023-03-29

**Confidence:** 5

**Summary Of Contributions:**

The paper suggests a method to improve the personalization performance of federated learning models, with the aim of optimizing the global model separately for local inference by each of the clients..

**Rating:**

Needs Clarification (NC): a submission which does not meet the reviewing criteria and needs clarification for its described problem or solution

**Strengths And Weaknesses:**

- While the work was relatively elucidated, the very few references suggest that a more thorough literature review can be pretty beneficial for the work.
- Personalization is the flagbearer of federated learning. FL started with the motivation to train and personalize models even further for users without having to share data. I believe the "problem" stated in this work is a standard approach in FL and not a "problem" for federated learning. FL's most significant issues are still along the lines of communication costs, computational costs, and fairness.

**Suggested Changes:**

As referenced in the Strengths/Weaknesses section, I believe that a more thorough review of current techniques is needed to ensure that there is a vacuum for this work direction to fill.

---

> ### Author Response · Authors · 2023-05-29
> **Response to Reviewer uWdB**
>
>  Thank you very much for your detailed feedback on our manuscript. We have revised our manuscript based on your comments. The responses to your specific comments are as following:
> >Summary Of Contributions: The paper suggests a method to improve the personalization performance of federated learning models, with the aim of optimizing the global model separately for local inference by each of the clients.
>
> >Comment: While the work was relatively elucidated, the very few references suggest that a more thorough literature review can be pretty beneficial for the work.
>
> Response: Thank you for your summary and suggestion. We really appreciate your efforts in reviewing our manuscript. We performed a more comprehensive literature review to indicate the gaps in the existing work and justify our contribution.
>
> >Comment: Personalization is the flagbearer of federated learning. FL started with the motivation to train and personalize models even further for users without having to share data. I believe the "problem" stated in this work is a standard approach in FL and not a "problem" for federated learning. FL's most significant issues are still along the lines of communication costs, computational costs, and fairness.
>
> Response: Thank you very much for this comment. The problem we address is that in personalized federated learning, the time to start personalization influences the accuracy of the final personalized model.
>
> >Comment: As referenced in the Strengths/Weaknesses section, I believe that a more thorough review of current techniques is needed to ensure that there is a vacuum for this work direction to fill.
>
> Response: Thank you for this comment. We present related work in the Introduction section to illustrate the current research gap. We have performed a more exhaustive literature review in the revised manuscript.
>
> Thank you very much for your valuable comment. Please let us know if there are any further suggestions that could improve the quality of our manuscript.

---

### Author Response · Authors · 2023-06-17
**Opting-in for Archival**

We wish to opt-in for archival

---

### Meta-Review · Area_Chair_PxQr · 2023-04-07

**Recommendation:** Invite to revise
**Confidence:** 3

**Metareview:**

Concerns regarding _reproducibility_ have been raised by reviewer `5Vou` (specifically, methodological details). Both reviewers `5Vou` and `uWdB` raise concerns regarding the specific contributions of the paper, which I would categorize under _clarity_. Consequently, I do not deem this paper to be CCR.


**Summary:**

To enhance personalization benefits in federated learning, the authors propose to begin the personalization of the client models before the global model has converged. In light of the reviewers' comments, this paper is not CCR.

**Comments And Feedback To The Authors:**

The paper is succinct which is great, but the writing can benefit from proofreading.

**Reason For Not Giving A Higher Recommendation:**

The authors should clarify whether similar work has done in the past and if so clearly delineate the distinctions of the current work. If the current proposal is a reimplementation of past work by others, this should be indicated.

Furthermore, as the reviewers point out, it would be very helpful if the authors could also clarify (by pointing to examples or showing the results of their own experiments) why the federated learning strategy that they describe as the dominant approach (personalization only after global model convergence) falls short of producing desirable levels of personalization, as claimed in the Introduction.

**Reason For Not Giving A Lower Recommendation:**

N/A

---

> ### Author Response · Authors · 2023-05-29
> **Response to Area Chair PxQr**
>
> Thank you so much for taking your time to review this manuscript. We really appreciate all your generous comments and suggestions. We have carefully studied the comments and made corrections to the manuscript, and our point-by-point responses are presented above.
> >Metareview: Concerns regarding reproducibility have been raised by reviewer 5Vou (specifically, methodological details). Both reviewers 5Vou and uWdB raise concerns regarding the specific contributions of the paper, which I would categorize under clarity. Consequently, I do not deem this paper to be CCR.
>
> >Summary: To enhance personalization benefits in federated learning, the authors propose to begin the personalization of the client models before the global model has converged. In light of the reviewers' comments, this paper is not CCR.
>
> Response: Thank you for your precious comments and advice. We took the following actions to address the issues raised by the reviewers and improve the overall quality of the paper:
> * We conducted a thorough review of the article, corrected grammatical errors and improved the writing quality.
> * We performed a more comprehensive literature review to indicate the gaps in the existing work and justify our contribution.
> * We have included a more detailed description of the proposed method to improve the clarity of the manuscript. And Table Algorithm 1 has been added to explain the flow of our proposed method.
> * We have updated the title to better indicate the focus of the manuscript.
>
> >Comment: The authors should clarify whether similar work has done in the past and if so clearly delineate the distinctions of the current work. If the current proposal is a reimplementation of past work by others, this should be indicated.
>
> Response: Thank you for pointing out the flaws in our manuscript. We present related work in the Introduction section to illustrate the current research gap. We asked the question ‘When will federated learning transfer from generalization to personalization?’ that currently has no corresponding work.
>
> >Comment: Furthermore, as the reviewers point out, it would be very helpful if the authors could also clarify (by pointing to examples or showing the results of their own experiments) why the federated learning strategy that they describe as the dominant approach (personalization only after global model convergence) falls short of producing desirable levels of personalization, as claimed in the Introduction.
>
> Response: Thank you for the suggestion. We illustrate the impact of the time to start personalization on the performance of federated learning models through examples in the Introduction. Starting personalization too early can cause important parameters of the global model to be changed while adapting to the client's local data, while too late can cause each client's final model to be misled by the global model.
>
> >Comment: The paper is succinct which is great, but the writing can benefit from proofreading.
>
> Response: Thank you for your careful review. We are very sorry for the mistakes in this manuscript and inconvenience they caused in your reading. We have thoroughly revised and rewritten the manuscript to improve its readability and clarity. We hope this improves the quality of the manuscript.
>
> We hope that the revised manuscript addresses your concerns. If there is anything else that needs improvement or clarification, please contact us.

---

### Decision · Program_Chairs · 2023-04-09

Revision accepted; invite to archive